# Hepatic vessel segmentation using a reduced filter 3D U-Net in ultrasound imaging

**Bart R. Thomson**[1,2]                                    BARTRTHOMSON@GMAIL.COM

[1] *Department of Technical Medicine, University of Twente, Enschede, The Netherlands*

[2] *Department of Surgical Oncology, The Netherlands Cancer Institute, Amsterdam, The Netherlands*

**Jasper Nijkamp**[2]                                        J.NIJKAMP@NKI.NL

**Oleksandra Ivashchenko**[2,3]                              O.IVASHCHENKO@NKI.NL

[3] *Department of Radiology, Leiden University Medical Center, Leiden, The Netherlands*

**Ferdinand van der Heijden**[1,2]                           F.VANDERHEIJDEN@UTWENTE.NL

**Jasper N. Smit**[2]                                        J.SMIT@NKI.NL

**Niels F.M. Kok**[2]                                        N.KOK@NKI.NL

**Koert F.D. Kuhlmann**[2]                                   K.KUHLMANN@NKI.NL

**Theo J.M. Ruers**[1,2]                                     T.RUERS@NKI.NL

**Matteo Fusaglia**[2]                                       M.FUSAGLIA@NKI.NL

## Abstract

Accurate hepatic vessel segmentation on ultrasound (US) images can be an important tool in the planning and execution of surgery, however proves to be a challenging task due to noise and speckle. Our method comprises a reduced filter 3D U-Net implementation to automatically detect hepatic vasculature in 3D US volumes. A comparison is made between volumes acquired with a 3D probe and stacked 2D US images based on electromagnetic tracking. Experiments are conducted on 67 scans, where 45 are used in training, 12 in validation and 10 in testing. This network architecture yields Dice scores of 0.740 and 0.781 for 3D and stacked 2D volumes respectively, comparing promising to literature and inter-observer performance (Dice = 0.879).

**Keywords:** Deep learning, Segmentation, Liver, Ultrasound, U-Net

## 1. Introduction

Surgical resections, when compared to other treatment plans, provide the best patient outcome for various types of liver malignancies (Kanas et al., 2012). Due to high complexity and inter-patient variability of underlying hepatic vascular anatomy, planning and execution of safe resection is challenging in surgery. Therefore, repetitive intraoperative imaging is required to monitor surgery progress and assess the tumour-vessel relationship in 3D. Currently, US is the only imaging modality that is widely accepted and integrated into a surgical workflow. Therefore, ultrasonography is the most suitable imaging modality for intraoperative visualisation of hepatic vasculature.

Despite many advantages of intraoperative ultrasound, it is still a primary 2D imaging modality, which complicates precise localization of each 2D image in 3D for a surgeon. An interactive visualization of automatically segmented vasculature in 3D would have been of

great value, yet challenging due to the complexity of US segmentation (Zhu et al., 2011; Noble and Boukerroui, 2006). In this work, an attempt to alleviate these challenges using 3D ultrasound imaging, in conjunction with vasculature segmentation has been proposed.

Other studies reported mean segmentation scores with a Dice of 0.5 (Wei et al., 2019) and an intersection over union (IoU) of 0.696 (Mishra et al., 2018) when segmenting vasculature on 2D images, using a 2D U-net and a simple convolutional neural network combined with k-means clustering respectively. 3D information has been shown to improve performance in biomedical volumetric segmentation (Çiçek et al., 2016) and can be acquired with a 3D US probe or by stacking 2D US images based on electromagnetic tracking.

In this study, a reduced filter 3D U-Net, chosen due to its popularity in medical image segmentation (Litjens et al., 2017), is proposed to achieve accurate vessel segmentation in true 3D (figure 1a) and stacked 2D (figure 1b & 1c) US images.

## 2. Materials and methods

The dataset contained 37 3D US scans and 30 2D acquired volumes, stacked based on electromagnetic tracking (Aurora Northern Digital — Ontario, Canada), data distribution is presented in table 1. Original 3D volume sizes were $512 \times 400 \times 256$ and stacked 2D volumes ranged from $293 \times 396 \times 526$ to $404 \times 572 \times 678$, depending on the zoom of the 2D slices, but were downsampled to 40% prior to training. All US imaging has been acquired intra-operatively in the NKI-AvL by 5 different observers. Each scan was delineated in 3D slicer (Kikinis et al., 2014) by one out of 4 annotators, annotations have been validated with an expert radiologist. To give a sense of scale of segmentation challenges, four scans were delineated by two observers, and Dice and IoU are reported as inter-observer variation.

The 3D U-Net architecture that is used is a NiftyNet (Gibson et al., 2018) Tensorflow implementation similar to Cicek et al. (Çiçek et al., 2016), however with an eighth of the amount of filters in every layer, to avoid bottlenecks. Adam optimizer, with learning rate $5 \times 10^{-3}$, L1 regularization with $10^{-5}$ weight decay, and a batch size of 2 was used. Training was performed on four NVIDIA 1080 GTX GPUs. Twenty patches with size $152 \times 152 \times 96$ were used per mean value normalized volume. All volumes were padded with a volume of $32 \times 32 \times 32$ and were augmented by rotating (between $-10°$ and $10°$), scaling (between $-10\%$ and $10\%$) and elastically deforming (S.D. 1). To reduce noise in the input images, a 3 pixel median filter was used to smooth the images. The Dice loss function was used in training of the network for 217 epochs which lasted 64 hours when there was no apparent converging of the validation loss. Segmentation accuracy was reported by means of Dice and IoU.

## 3. Experiment and Results

The reduced filter U-Net obtained Dice scores of 0.740 ($\pm 0.02$) and 0.781 ($\pm 0.07$) for the 3D acquired and 2D stacked US images respectively compared to an inter-observer variability of 0.879 ($\pm 0.02$) (table 1). IoU is reported at 0.584 ($\pm 0.02$), 0.645 ($\pm 0.09$) and 0.785 ($\pm 0.03$) respectively. Figure 1 shows segmentation results for several selected cases.

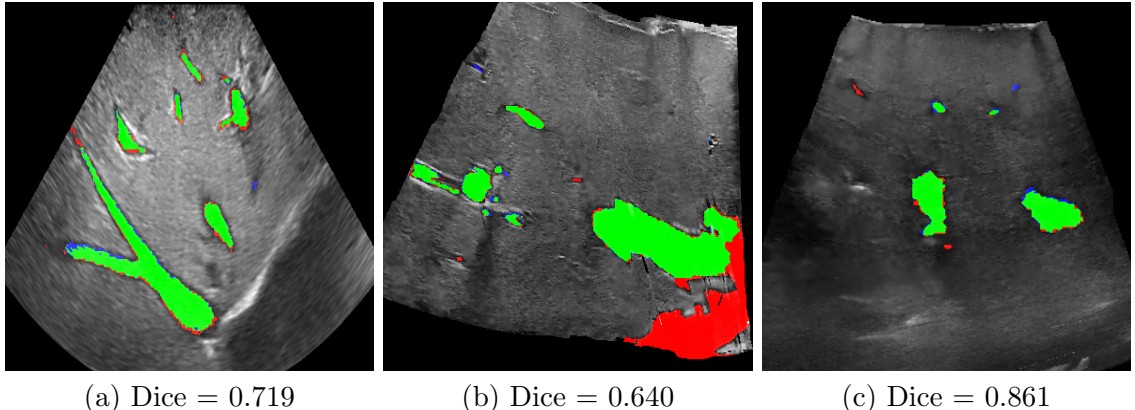

(a) Dice = 0.719          (b) Dice = 0.640          (c) Dice = 0.861

Figure 1: Examples of test set segmentation results, true positives are colored green, false positives blue and false negatives red

Table 1: Comparison of Dice scores for vessel segmentation in true 3D US, stacked 2D and inter-observer

| US modality | Mean Dice | Mean IoU | Training | Validation | Test | Total |
|---|---|---|---|---|---|---|
| 3D | 0.740 ±0.02 | 0.584 ±0.02 | 27 | 7 | 3 | 37 |
| 2D | 0.781 ±0.07 | 0.645 ±0.09 | 18 | 5 | 7 | 30 |
| Inter-observer | 0.879 ±0.02 | 0.785 ±0.03 | | | 4 | 4 |

## 4. Discussion and Conclusion

This study shows that it is possible to accurately segment hepatic vessels on US imaging with a relatively small dataset, but deviates from the inter-observer performance. Furthermore, our results seem favorable (Milko et al., 2008; Wei et al., 2019), but also slightly underperform (Mishra et al., 2018) when compared to 2D segmentation literature. Overall under-segmentation of the inferior vena cava is observed, especially near the edges of the volume (figure 1b). We suspect that this is caused by incomplete vessel information (i.e. incomplete visibility of vessel cross section), strongly influencing the Dice due to its large volume. Comparing the proposed network to a network with the original amount of filters was not possible due to GPU memory limitations. The learned features between the different acquisition methods appear exchangeable as there appears no difference in segmentation performance. Whilst promising results have been demonstrated, further validation will be done by expanding the data set. In the future we will expand this methodology by discriminating between different types of vasculature (i.e. hepatic and portal vein) as well as parenchyma. Moreover, we will explore the use of these segmentations for automatic registration with a MRI model in a navigation surgery setting. These segmentations are expectantly sufficient to realize a centerline-based registration pipeline in the future.

In conclusion we demonstrate that a 3D U-Net architecture with a reduced amount of filters is able to accurately segment hepatic vasculature.

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
