# OpenReview forum: "Hepatic vessel segmentation using a reduced filter 3D U-Net in ultrasound imaging"
_MIDL.io/2019/Conference/Abstract — MIDL Abstract 2019_

### Official Review · AnonReviewer1 · 2019-04-29
**ultrasound segmentation via DL**

**Rating:** 2
**Confidence:** 3

**Review:**

The authors propose to segment hepatic vessels using a 3D or a 2D fully convolutional neural network trained on annotated data depicting the liver in ultrasound. the work uses a rather standard technique and achieves binary segmentation of the vessels present in the image. the authors sufficiently describe the methodology and the motivation of the proposed work. The algorithm used in this study is very standard and not much different from other algorithms used in similar studies in a wide range of modalities. the discrete results achieved by this method do not come as a surprise.

The dataset used in this study does not seem to be publicly available. Moreover the results seem not to match very closely the inter rater observed variability on a fraction of the same dataset. The fact that 2D and 3D FCNN trained with Dice loss work for such a problem, delivering discrete results, is not a surprise. It's surprising that the performance in 3D are actually lower than 2D. This might be due to the patch based methodology that needed to be adopted in 3D due to the limitation of the GPU used in this study and in general memory issues that are often encountered in 3D.
As the ground truth was obtained directly in 3D via Slicer, the lower performance in 3D are even less justified.

I think the work should be developed a little more and re-submitted at a later stage.

---

### Official Review · AnonReviewer2 · 2019-04-29
**Not a novel method, but a nice direction of research**

**Rating:** 3
**Confidence:** 2

**Review:**

The authors propose to use a reduced-filter 3-D U-net architecture to detect hepatic vasculature in 3D US volumes. They achieve a Dice score of 0.740 and 0.781 for 3D and stacked 2D volumes respectively.

Pros

- The proposed method and specifics of the training are well-explained.
- Promising results are achieved and inter-observer performance is available for comparison.

Cons

- The utilized method is not a novel architecture and authors do not specify the specifics of hepatic vessel segmentation in US images in relation to the selected architecture and training objective.

Minor issues/suggestions
- The authors can consider a novel loss function to suppress the high number of true positives (e.g. Fig. 1b)
- Please use capital letter for Figures and Tables (e.g. Figure 1a)

---

### Decision · Program_Chairs · 2019-05-06
**Acceptance Decision**

Accept